# Unifying Long and Short Spatio-Temporal Forecasting with Spectral Graph Neural Networks

## Abstract

Multivariate Time Series (MTS) forecasting plays a vital role in various practical applications. Current research in this area is categorized into Spatial-Temporal Forecasting (STF) and Long-term Time Series Forecasting (LTSF). While these tasks share similarities, the methods and benchmarks used differ significantly. Spatio-Temporal Graph Neural Networks (STGNNs) excel at modeling interrelationships in STF tasks but face difficulties with long sequence inputs due to inefficient training. In contrast, LTSF models handle long sequences well but struggle with capturing complex variable interrelationships. This paper proposes the Spectral Spatio-Temporal Graph Neural Network (S2GNN) to address these challenges, unifying short- and long-sequence spatio-temporal forecasting within a single framework. S2GNN leverages a decoupled GNN along with an MLP architecture to ensure efficiency. Specifically, it employs spectral GNNs for global feature extraction on an adaptive graph structure and uses MLP to process multiple feature embeddings, enabling it to handle varying sequence lengths. Additionally, we introduce scale-adaptive node embeddings and cross-correlation embeddings for better differentiation between similar temporal patterns. Extensive experiments on eight public datasets, including both STF and LTSF datasets, demonstrate that S2GNN consistently outperforms state-of-the-art models across diverse prediction tasks. Code is available at `https://anonymous.4open.science/r/S2GNN-B21D`.

## 1 Introduction

Current Multivariate Time Series (MTS) prediction methods can be broadly categorized into Spatio-Temporal Forecasting (STF) and Long-term Time Series Forecasting (LTSF) (Shao et al., 2023). STF focuses on capturing both spatial and temporal dependencies, with Spatio-Temporal Graph Neural Networks (STGNNs) being a prominent approach (Jiang et al., 2023). These models are commonly evaluated on spatio-temporal datasets, such as those from the PEMS (Chen et al., 2001) traffic series. In contrast, LTSF emphasizes learning patterns like seasonality or trends over longer input sequences and utilizes long-sequence datasets that cover a wide range of scenarios to ensure robust model generalization (Qiu et al., 2024; Zhou et al., 2021).

The key distinction between these two research areas lies in their ability to process input sequences. STF models are typically designed for short-term inputs and may struggle with longer sequences, partly due to the high training costs associated with some models (Han et al., 2024). On the other hand, LTSF models are better suited for handling longer input sequences but often face challenges in capturing spatio-temporal relationships and tend to incur high computational complexity, especially when using transformer-based architectures (Huang et al., 2023). Figure 1 shows the performance of several STF models on short-term and long-term inputs. As the figure indicates, the FLOPs of most models increase significantly with input length, making long-term training nearly impossible. Thus, there is a clear need for an STGNN model capable of handling both short and long input sequences efficiently.

As a crucial part of STGNN models, most existing approaches rely on GCN-based methods (Kipf & Welling, 2016). However, current research in GNNs mainly follows two directions: spatial-based

and spectral-based approaches (Guo & Wei, 2023). Spatial GNNs capture node representations through message passing and aggregation among neighboring nodes, often acting as low-pass filters (Wu et al., 2019a). In contrast, spectral-based GNNs perform convolutions in the spectral domain of the graph Laplacian, offering more flexibility in handling different frequency components (He et al., 2021). In dynamic or evolving graphs, spectral GNNs may be less robust due to their reliance on the graph Laplacian, which is sensitive to changes in the graph structure. As a result, spatial GNNs are often employed in existing work to learn spatial relationships between variables (Zheng et al., 2024), though we explore whether this assumption holds universally in this paper.

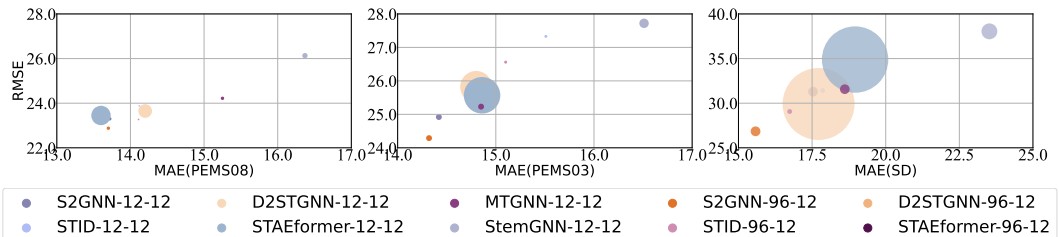

Figure 1: Comparison of existing models on short-term STF and long-term STF results. From left to right, the number of nodes increases across three datasets. The size of the circles represents the FLOPs required to train for one epoch. Existing models struggle to balance prediction accuracy and training efficiency when handling long-term STF tasks.

To address the limitation of static graph structures in capturing dynamic changes in nodes, as well as the use of spatiotemporal models in scenarios without graph structures, adaptive graph structures are widely used in STGNNs and are typically assumed to change continuously throughout training, making spectral GNNs seem impractical for dynamic environments due to their sensitivity to graph structure changes. This raises an interesting question: how much do adaptive graph structures, often constructed using randomly initialized node embedding vectors, actually change during training? Figure 2 illustrates the training process of two existing models that use adaptive graph structures. Evidence across various datasets shows that, contrary to common belief, both the connectivity and numerical variations of the adaptive graphs stabilize gradually in training. This observation challenges the assumption that spectral GNNs are too sensitive and motivates the use of spectral GNNs with learnable filters on adaptive graph structures. Additionally, from the perspective of perturbation theory for eigenvectors, if the change in the corresponding eigenvalue is sufficiently small, the eigenvector will not shift significantly (Spielman, 2019).

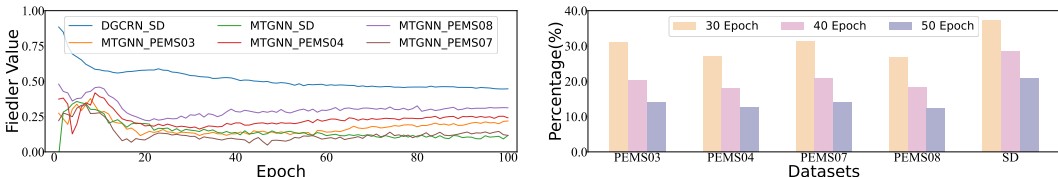

Figure 2: Observations of adaptive graph structure in STF datasets. **Left:** The changes in the connectivity of the graph structure tend to stabilize as the number of epochs increases. **Right:** The difference in the percentage of Frobenius norm between the best epoch and the 30th, 40th, and 50th epochs, respectively.

Meanwhile, for the sake of training efficiency, we adopt a simple yet effective model architecture for the sequential modelling. We review the concept of sample indistinguishability, which first proposed in (Deng et al., 2021). Although previous work has addressed the indistinguishability in the temporal and spatial domains separately (Shao et al., 2022a), we believe that the current approaches still have certain limitations. Firstly, existing methods for handling spatial distinguishability cannot adapt to changes in the number of nodes. Learning node representations in a fixed $d$-dimensional feature space becomes more challenging as the number of nodes increases. Secondly, embedding methods using temporal information are influenced by the sequence's periodicity, and incomplete temporal

data can reduce the model's ability to distinguish patterns. Thus, an embedding method that directly learns distinguishable features from the input sequence is needed.

To address the first issue, we use sparse initialization for node embeddings, leveraging the Restricted Isometry Property (RIP) theory (Tropp, 2017) to link sparsity with the number of nodes, allowing the embeddings to adapt to changes in node scale. For the second issue, we compute cross-correlation between the input and feature sequences to learn features adaptively. This dynamic addition of feature embedding vectors enhances the model's ability to resolve sample indistinguishability.

**Our contributions.** In summary, our main contributions are as follows:

- We propose the Spectral Spatio-Temporal Graph Neural Network (S2GNN), which incorporates learnable filters capable of adaptively capturing and processing both short- and long-sequence features, enhancing the model's expressiveness and versatility. The decoupled GNN along with an MLP architecture ensures overall efficiency.

- We introduce a scale-adaptive node embedding method that leverages sparsity to accommodate varying node scales and a cross-correlation embedding method that adaptively learns to distinguish features from the input sequences.

- Experimental results reveal that both low-pass and high-pass information must be considered when using learnable filters in conjunction with the adaptive graph. Meanwhile, our model balances training efficiency and performance, achieving superior results on eight public datasets compared to existing models.

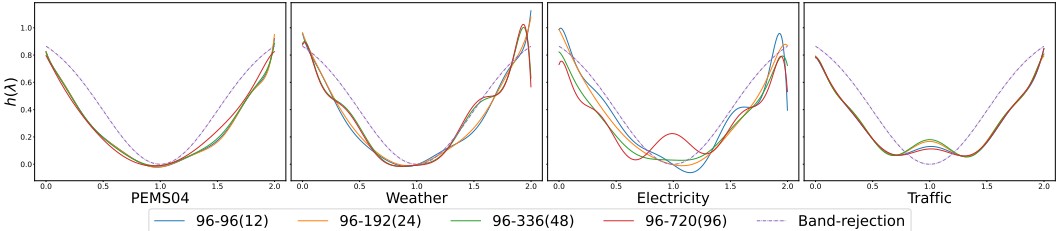

Figure 3: Visualization of learned graph filters. The results show that the filters of the adaptive graph resemble a band-rejection filter across all public datasets, indicating the presence of both homophilic and heterophilic relationships between nodes.

## 2 PRELIMINARIES

### 2.1 SPECTRAL GRAPH NEURAL NETWORK

Spectral GNNs work in the frequency (spectral) domain using graph Laplacian, which enables them to apply various filters to the graph signal (node features) (Bruna et al., 2013). The graph filters could be fixed (Kipf & Welling, 2016) (Klicpera et al., 2018) (Wu et al., 2019a) or approximated with polynomials (Defferrard et al., 2016) (Chien et al., 2020) (He et al., 2021). Spectral graph models naturally excel at learning heterophilic relationships. Since the homophilic/heterophilic relationships in adaptive graphs are uncertain, spectral models are well-suited for use on adaptive graphs if we can ignore their dynamic changes. Among existing spectral GNNs, **ChebNetII** (He et al., 2022) can approximate an arbitrary spectral filter $h(\lambda)$ with an optimal convergence rate, and the learnable parameter $\gamma_j$ allows the model to learn an arbitrary spectral filter via gradient descent. Additionally, it offers strong interpretability. It is formulated as:

$$\mathbf{Y} = \frac{2}{K+1} \sum_{k=0}^{K} \sum_{j=0}^{K} \gamma_j T_k(x_j) T_k(\hat{\mathbf{L}}) f_\theta(\mathbf{X}) \tag{1}$$

where $x_j = cos((j+1/2)\pi/(K+1))$ and the Chebyshev nodes of $T_{K+1}$, $f_\theta(\mathbf{X})$ denotes an MLP on the node feature matrix $\mathbf{X}$, and $\gamma_j$ for $j = 0, 1 \cdots, K$ are the learnable parameters.

## 2.2 Multivariate Time Series Forecasting

Multivariate time series forecasting covers a wide range of methods. Solely from the perspective of explicitly leveraging inter-variable relationships, models can be broadly categorized into GNN-based models, cross-variable interaction models and cross-time interaction models. First, we provide some definitions to facilitate a quicker understanding of the relevant concepts.

**Time series.** A time series $\mathbf{X} \in \mathbb{R}^{N \times T}$ is a time oriented sequence of $N$-dimensional time points, where $T$ is the number of time points, and $N$ is the number of variables. When $N = 1$, a time series is called univariate. When $N > 1$, it is called multivariate.

**Spatial-Temporal Forecasting (STF)** primarily focuses on modeling the temporal and spatial patterns of multivariate time series. Let $G = (\mathcal{V}, \mathcal{E})$ be a network as an undirected graph with $V = |\mathcal{V}|$ nodes and $E = |\mathcal{E}|$ edges. It involves learning a model $f$ that, given observations of historical look-back observations $\mathbf{X} \in \mathbb{R}^{V \times P}$ of $P$ timesteps on a graph $G$, outputs predictions $\mathbf{Y} \in \mathbb{R}^{V \times F}$ for the future horizon of length $F$. The process is defined as:

$$[x_{t-P+1}, \cdots, x_{t-1}, x_t; G] \xrightarrow{f_\theta} [x_{t+1}, x_{t+2}, \cdots, x_{t+F}]$$

where $x_t \in \mathbb{R}^V$ is a graph signal vector at time step $t$, $\theta$ is a collection of learnable parameters. In this paper, the number of nodes is equal to the number of variables, i.e., $N = V$.

**Long-term Time Series Forecasting (LTSF)** primarily focus on modeling the long-term dependencies of multivariate time series. The definition of LTSF is similar to STF, except that it does not explicitly use a graph structure and typically involves a longer input data sequence.

With the development of GCN (Defferrard et al., 2016) (Kipf & Welling, 2016), models begin to utilize GCNs to model spatial dependencies based on a pre-defined prior graph and further combine them with sequential models (Yu et al., 2017) (Li et al., 2017) (Wu et al., 2019b). However, many recent works argue that the pre-defined prior graph might be biased, incorrect, or even unavailable in many cases. Thus, they propose to jointly learn the graph structure (i.e., adaptive graph) (Bai et al., 2020) (Jin et al., 2022) (Shang et al., 2021) or use graph-free methods to do STF tasks (Shao et al., 2022a) (Liu et al., 2023b) (Deng et al., 2021).

Cross-variable interaction models are also critical for time series forecasting. Recently, (Zhang & Yan, 2023) and (Liu et al., 2023c) both adopted channel-wise Transformer-based frameworks, and extensive experimental results have demonstrated the effectiveness of channel-wise attention for time series forecasting.

Cross-time interaction models focus on extracting information along the time dimension, such as patch-wise patterns (Nie et al., 2022), multi-scale information (Wang et al., 2024) (Shabani et al., 2022), or frequency-domain features (Wu et al., 2021) (Piao et al., 2024).

## 3 Methodology

In this section, we first briefly review the concept of sample indistinguishability, and then we present our proposed embedding methods. Next, we introduce the adaptive graph structure widely used in the field of STF, as well as how graph filters are applied to this structure. Finally, we provide an overall illustration of the S2GNN architecture and the prediction module, which consists of multiple MLPs.

### 3.1 Indistinguishability of Times Series Samples

Simply speaking, sample indistinguishability occurs when dynamics from different factors overlap, making input sequence features too similar to distinguish. (Deng et al., 2021) points out that this issue hinders deep neural networks from capturing spatial and temporal differences, while (Shao et al., 2022a) offers a simple framework to mitigate it. In this paper, we introduce a scale-adaptive node embedding method to adapt the model to varying node numbers and a cross-correlation embedding method to extract distinguishable features directly from the input sequence, addressing cases where temporal information is unavailable or ineffective.

**Scale-Adaptive Node Embeddings.** (Yeh et al., 2024) shows that random projection can enhance a model's ability to capture spatial relationships within input sequences. In our work, we leverage this

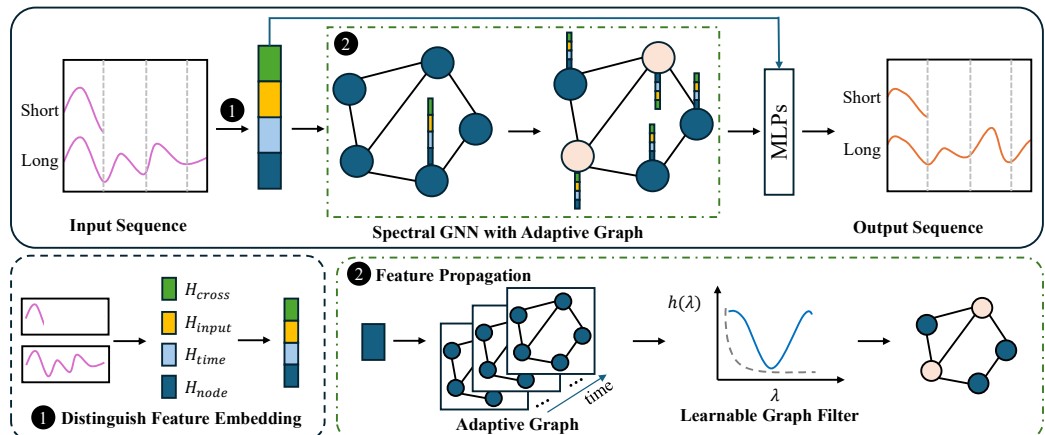

Figure 4: Overall architecture. We first extract distinguishable features from the input sequence and then propagate these features on an adaptive graph while learning filters suited to the adaptive graph. Finally, we use multi-layer MLPs for learning. The overall structure of S2GNN is simple yet effective, allowing it to handle both short-term and long-term STF tasks.

approach along with sparse initialization to generate node embeddings. The benefit of this method is that under the Restricted Isometry Property (RIP) (Tropp, 2017) theory, we can use sparsity to link the number of nodes with the node embeddings. It allows the dimensionality of the node embeddings used in training to remain within a smaller range ($h << N$), bypassing the increased learning difficulty caused by a larger number of nodes. Assuming $r \in (0,1)$ represents sparsity, the dimension $h$ of the random projected matrix should satisfy:

$$h = c \cdot r \log(N/r). \tag{2}$$

Where $c$ is a constant, $N$ represents the number of nodes. Therefore, our Scale-Adaptive Node Embeddings is formulated as following:

$$
\begin{aligned}
\hat{\mathbf{W}}_{\mathbf{node}} &= \text{Linear}_1(\text{Linear}_2(\mathbf{W_{node}}^T))^T, \\
\mathbf{H_{node}} &= \hat{\mathbf{W}}_{\mathbf{node}}[\mathbf{X_{node}}],
\end{aligned}
\tag{3}
$$

where $W_{node}$ and $\hat{W}_{node} \in \mathbb{R}^{N \times d}$ and $\text{Linear}_2$ refers to fixed random peojection operation. It is noticed that our randomly initialized node embeddings $W_{node}$ are fixed and do not participate in training process, which is different from existing methods. $X_{node}$ represents node ID in the input sequence.

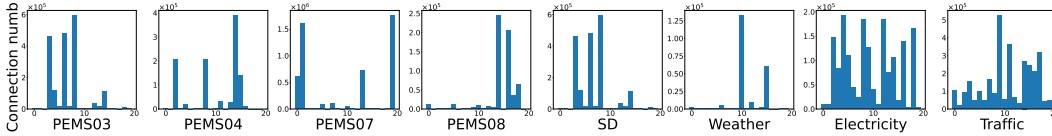

Figure 5: The connection number of each kernel in each dataset. The distribution demonstrates that the cross-correlation embedding method effectively distinguishes the input sequences.

**Cross-Correlation Embeddings.** The key to alleviating sample indistinguishability is to map a certain feature of the input sequence to an embedding vector using a meaningful value. This value could be the node's ID, the time of the day, or the day of the week. To avoid relying on external information, we calculate the cross-correlation between the input sequence and a set of feature sequences. The input sequence is then connected to embedding vectors based on the magnitude of the cross-correlation. The calculation process can be efficiently achieved using convolution operation, as it inherently performs cross-correlation operation.

$$\mathbf{C_t} = \text{Cross-Correlation}(\mathbf{X_{t-L+1:t}}, k_i),$$
$$\mathbf{H_{cross}} = \mathbf{W_{cross}}[\text{argmax}(\mathbf{C_t})], \tag{4}$$

where $k_i$ refer to the kernel in convolution. The index corresponding to the maximum value from multiple convolution operations is then selected as the index for the corresponding feature embedding matrix.

A more complex convolutional neural network could be designed to learn distinguishable features from the input sequence as long as it produces a value for indexing. However, for simplicity, we use a convolution kernel with the same length as the input sequence so that we can directly validate the cross-correlation embedding method and visualize the kernel to see the distinguishing features. Figure 5 shows the number of times each feature sequence is selected in the test set. Multiple feature sequences are chosen across different datasets, confirming the effectiveness of our method. Figure 6 shows the learned distinguishable features, where the color intensity of each sequence corresponds to the number of connections. The diversity of these features is clearly visible.

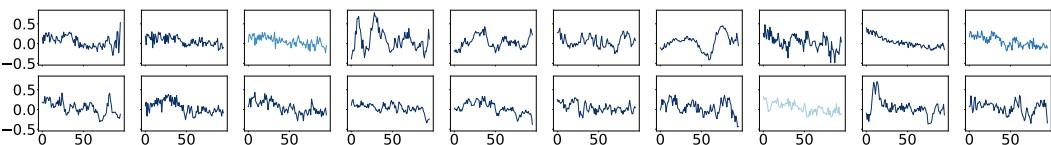

Figure 6: A visualization of the convolutional kernels used in the cross-correlation embedding process, where the color intensity represents their usage frequency (i.e., the number of times each kernel is connected).

### 3.2 ADAPTIVE ADJACENT MATRIX IN TIME SERIES

The term "adaptive graph structure" generally refers to any adjacency matrix derived from the node embeddings. In existing works, it is common to initialize a set of node embeddings as source nodes and another set of node embeddings as target nodes, followed by employing various computation methods to obtain an adjacency matrix that can be adapted during training. In this paper, we adopt the most straightforward approach: we directly represent each node's embedding with a set of trainable weights and compute the cosine similarity between node embeddings as the dynamic adjacency matrix. The calculation of the adjacent matrix and its Laplacian matrix are as follows:

$$\mathbf{A} = \text{ReLU}(\text{CosineSimilarity}(\hat{\mathbf{W}}_{\mathbf{node}}))$$
$$\mathbf{L} = \mathbf{I} - \mathbf{D}^{-1/2}\mathbf{A}\mathbf{D}^{-1/2} \tag{5}$$
$$\hat{\mathbf{L}} = 2\mathbf{L}/\lambda_{max} - \mathbf{I},$$

where $\lambda_{max}$ is the largest eigenvalue of $\mathbf{L}$. Suppose we have the distinguishable features $\mathbf{H}$, then the propogation process on the adaptive graph is formulated as:

$$\mathbf{Z} = \frac{2}{K+1}\sum_{k=0}^{K}\sum_{j=0}^{K}\gamma_j T_k(x_j)T_k(\hat{\mathbf{L}})\mathbf{H}, \tag{6}$$

where $\gamma_j$ for $j = 0, 1 \cdots, K$ are the learnable parameters. $x_j = cos((j + 1/2)\pi/(K + 1))$ and the Chebyshev nodes of $T_{K+1}$, we set $K = 10$ as a default hyperparameter.

### 3.3 OVERALL STRUCTURE

In total, we have multiple feature embeddings concatenated altogether as distinguishing feature embeddings. Moreover, we concatenate the propagated feature embeddings with the original ones. This approach is as follows:

$$\mathbf{H} = \mathbf{H_{input}} \parallel \mathbf{H_{node}} \parallel \mathbf{H_{time}} \parallel \mathbf{H_{cross}}$$
$$\mathbf{Z} = \text{SpectralGNN}(\mathbf{H}) \tag{7}$$
$$\hat{\mathbf{Z}} = \text{Concat}(\mathbf{H}, \text{Linear}(\mathbf{Z})).$$

We use $\hat{\mathbf{Z}}$ obtained from above as the input for the prediction module. The overall structure of the prediction module is a stack of MLPs, each with a residual connection. In the end, we obtained the prediction results of another linear regression on the last layer. We use the mean absolute error (MAE) as the loss function on spatialtemporal datasets and the mean squared error (MSE) on long-sequence datasets:

$$\hat{\mathbf{Z}}^{\mathbf{l+1}} = f_\theta(\hat{\mathbf{Z}}^{\mathbf{l}}) + \hat{\mathbf{Z}}^{\mathbf{l}}$$
$$\hat{\mathbf{Y}} = \text{Linear}(\hat{\mathbf{Z}}^{\mathbf{last}}). \tag{8}$$

## 4 EXPERIMENTS

In this section, our primary goal is to demonstrate the effectiveness and feasibility of S2GNN in handling both long and short sequence inputs. We first introduce the basic experimental setup, followed by presenting the results of S2GNN on real-world data. Afterward, we compare the computational cost of training S2GNN and baseline models for one epoch across various datasets. Although different models require varying total epochs for training and have different memory demands, the comparison of FLOPs still provides an intuitive reflection of the challenges faced during model training. Finally, we visualized the prediction results of S2GNN and the baseline models on certain datasets. From these visualizations, we were able to identify some of the reasons why existing models fail in specific scenarios.

### 4.1 EXPERIMENT SETTINGS

**Datasets.** We conducted experiments on eight real-world datasets, including five traffic flow datasets commonly used in spatio-temporal forecasting tasks (PEMS03, PEMS04, PEMS07, PEMS08, and SD) and three benchmark datasets typically employed for long-term time series forecasting tasks (Electricity, Weather, and Traffic).

**Baselines.** We selected several existing spatiotemporal models and compared their performance on both short-term and long-term inputs in Figure 1. Among these, D2STGNN (Shao et al., 2022b) require pre-defined graph structures, while StemGNN (Cao et al., 2020) and MTGNN (Wu et al., 2020) employs an adaptive graph structure. Although STID (Shao et al., 2022a) and STAEformer (Liu et al., 2023a) do not utilize graph structures, they have demonstrated effectiveness in short-term STF tasks. Detailed results are in the appendix (Table 3).

For the LTSF tasks, we selected various types of models from the perspective of explicitly leveraging inter-variable relationships as benchmark baselines. DLinear (Zeng et al., 2022) is one of the simplest and most effective recent models. PatchTST (Nie et al., 2022) processes time series data by segmenting it into patches, which is widely adopted nowadays. MTGNN (Wu et al., 2020) has proven effective for both STF and LTSF. TimesNet (Wu et al., 2022) and iTransformer (Liu et al., 2023c) focus on leveraging relationships between variables for forecasting. CrossGNN (Huang et al., 2023) is a recently proposed spatial GNN model with linear complexity that balances both efficiency and predictive performance in LTSF tasks.

**Evaluation metrics.** Following previous work, for STF tasks, we compared the MAE, RMSE, and MAPE using the re-normalized prediction results, benchmarking against other models. For LTSF tasks, we used the normalized prediction results and compared the models based on MAE and MSE. To simplify the presentation of results across both STF and LTSF datasets, we omitted MAPE from the main text since the other two metrics used are sufficient to support the conclusions of the paper.

**Implementation details.** Our experiments were conducted on a 16GB V100 and a 40GB NVIDIA A100 GPU, averaging results over three random seeds. Due to the training efficiency of existing models and our computational resource limits, we reproduced only a subset of them. Our reproduced results on long-sequence data were slightly worse than those in the CrossGNN paper, so we

used their reported results for comparison. Most of our computational resources focused on the spatiotemporal datasets. For the spatiotemporal forecasting (STF) task, we used two settings: one with an input length of $P = 12$ to predict the next $F = 12$ steps and another with $P = 96$ to predict future outputs at $F = \{12, 24, 48, 96\}$ steps. For LTSF, we used $P = 96$ to predict future results at $F = \{96, 192, 336, 720\}$. We used AdamW as the optimizer with a learning rate of 0.005 and trained each model for 100 epochs on each dataset, with a batch size of 64.

Table 1: Experimental results on long-term STF datasets and LTSF datasets.

| Methods | Ours | | iTransformer | | CrossGNN | | TimesNet | | PatchTST | | DLinear | | MTGNN | |
|---|---|---|---|---|---|---|---|---|---|---|---|---|---|---|
| Metric | MAE | RMSE | MAE | RMSE | MAE | RMSE | MAE | RMSE | MAE | RMSE | MAE | RMSE | MAE | RMSE |
| PEMS03 12 | **14.32** | 24.29 | 15.60 | 26.70 | 17.46 | 27.56 | 17.26 | 27.68 | 17.11 | 27.20 | 20.83 | 33.99 | 14.64 | **23.91** |
| PEMS03 24 | **16.08** | **27.49** | 18.03 | 30.47 | 22.34 | 35.41 | 19.66 | 31.65 | 21.21 | 34.09 | 29.50 | 49.12 | 66.62 | 101.44 |
| PEMS03 48 | **18.90** | **32.95** | 22.36 | 37.80 | 30.55 | 47.48 | 24.24 | 40.40 | 28.12 | 45.63 | 46.00 | 74.96 | 61.21 | 95.70 |
| PEMS03 96 | **21.24** | **36.96** | 26.41 | 44.12 | 41.98 | 62.41 | 30.18 | 50.73 | 36.50 | 58.39 | 69.01 | 103.58 | 82.78 | 114.90 |
| PEMS04 12 | **18.16** | **29.74** | 20.31 | 32.43 | 23.84 | 37.31 | 21.48 | 33.80 | 26.08 | 40.80 | 27.06 | 42.75 | 18.79 | 30.55 |
| PEMS04 24 | **19.21** | **31.28** | 22.63 | 35.84 | 30.26 | 46.93 | 23.57 | 36.52 | 33.47 | 52.27 | 37.10 | 58.58 | 73.56 | 113.67 |
| PEMS04 48 | **20.68** | **33.47** | 26.03 | 40.93 | 42.15 | 63.62 | 26.93 | 41.65 | 47.73 | 73.46 | 56.16 | 87.05 | 52.26 | 89.81 |
| PEMS04 96 | **21.47** | **34.63** | 30.35 | 47.58 | 55.69 | 80.88 | 31.54 | 48.45 | 63.75 | 96.75 | 82.47 | 120.03 | 90.49 | 128.65 |
| PEMS07 12 | **19.10** | **32.21** | 21.55 | 35.38 | 25.97 | 40.64 | 25.15 | 41.57 | 28.49 | 44.20 | 30.97 | 48.50 | 20.81 | 33.83 |
| PEMS07 24 | **20.76** | **34.87** | 24.71 | 40.50 | 33.76 | 51.55 | 28.30 | 40.09 | 37.03 | 57.61 | 44.42 | 69.55 | 22.41 | 36.90 |
| PEMS07 48 | **22.96** | **38.36** | 29.05 | 47.53 | 46.59 | 68.79 | 32.19 | 53.10 | 52.19 | 81.34 | 69.99 | 105.61 | 62.66 | 105.80 |
| PEMS07 96 | **23.70** | **40.50** | 34.19 | 54.78 | 60.42 | 85.12 | 37.41 | 62.42 | 65.03 | 102.00 | 105.04 | 143.06 | 109.37 | 151.35 |
| PEMS08 12 | **13.70** | **22.88** | 15.15 | 24.73 | 18.62 | 29.52 | 19.08 | 30.82 | 20.60 | 32.29 | 21.60 | 34.37 | 14.93 | 23.89 |
| PEMS08 24 | **14.87** | **24.88** | 17.00 | 27.96 | 23.94 | 37.43 | 21.87 | 34.66 | 26.87 | 42.14 | 30.39 | 48.35 | 67.91 | 104.51 |
| PEMS08 48 | **16.17** | **26.87** | 19.99 | 32.94 | 32.55 | 49.32 | 27.13 | 42.44 | 37.75 | 58.65 | 48.37 | 74.35 | 47.88 | 82.05 |
| PEMS08 96 | **17.36** | **28.88** | 23.35 | 38.27 | 44.14 | 64.15 | 30.01 | 47.99 | 50.58 | 79.10 | 75.35 | 106.21 | 84.43 | 118.62 |
| SD 12 | **15.58** | 26.85 | 17.30 | 29.07 | 23.44 | 39.19 | 19.64 | 32.50 | 23.09 | 37.82 | 28.26 | 49.85 | 17.11 | 29.50 |
| SD 24 | **17.96** | **32.19** | 19.80 | 33.98 | 29.65 | 50.82 | 23.07 | 38.17 | 26.18 | 44.32 | 33.45 | 59.85 | 19.77 | 34.74 |
| SD 48 | **20.32** | **37.78** | 22.40 | 38.65 | 34.22 | 58.41 | 23.35 | 39.33 | 27.06 | 45.69 | 36.61 | 65.24 | 59.00 | 101.37 |
| SD 96 | **21.82** | **40.65** | 24.68 | 43.81 | 37.60 | 65.39 | 24.84 | 42.19 | 30.21 | 50.40 | 38.37 | 67.92 | 104.00 | 144.55 |
| Metric | MAE | MSE | MAE | MSE | MAE | MSE | MAE | MSE | MAE | MSE | MAE | MSE | MAE | MSE |
| Weather 96 | **0.198** | **0.154** | 0.211 | 0.175 | 0.218 | 0.159 | 0.220 | 0.172 | 0.230 | 0.171 | 0.255 | 0.196 | 0.329 | 0.230 |
| Weather 192 | **0.243** | **0.200** | 0.253 | 0.210 | 0.266 | 0.211 | 0.261 | 0.219 | 0.271 | 0.219 | 0.296 | 0.237 | 0.322 | 0.263 |
| Weather 336 | 0.304 | 0.277 | 0.295 | 0.268 | 0.310 | **0.267** | 0.306 | 0.280 | 0.321 | **0.277** | 0.335 | 0.283 | 0.396 | 0.354 |
| Weather 720 | 0.363 | 0.366 | **0.351** | **0.343** | 0.362 | 0.352 | 0.359 | 0.365 | 0.367 | 0.365 | 0.381 | 0.345 | 0.371 | 0.409 |
| Electricity 96 | **0.226** | **0.135** | 0.237 | 0.147 | 0.275 | 0.173 | 0.272 | 0.168 | 0.268 | 0.159 | 0.276 | 0.194 | 0.318 | 0.217 |
| Electricity 192 | **0.244** | **0.154** | 0.255 | 0.166 | 0.288 | 0.195 | 0.289 | 0.184 | 0.278 | 0.177 | 0.280 | 0.193 | 0.352 | 0.238 |
| Electricity 336 | **0.264** | **0.172** | 0.277 | 0.187 | 0.300 | 0.206 | 0.300 | 0.198 | 0.296 | 0.195 | 0.296 | 0.206 | 0.348 | 0.260 |
| Electricity 720 | 0.305 | 0.215 | **0.299** | **0.214** | 0.335 | 0.231 | 0.320 | 0.220 | 0.317 | 0.215 | 0.329 | 0.242 | 0.369 | 0.290 |
| Traffic 96 | **0.251** | 0.508 | 0.260 | **0.404** | 0.310 | 0.570 | 0.321 | 0.593 | 0.319 | 0.583 | 0.396 | 0.650 | 0.437 | 0.660 |
| Traffic 192 | **0.263** | 0.538 | 0.269 | **0.428** | 0.321 | 0.577 | 0.336 | 0.617 | 0.331 | 0.591 | 0.370 | 0.598 | 0.438 | 0.649 |
| Traffic 336 | **0.272** | 0.567 | 0.293 | **0.433** | 0.324 | 0.588 | 0.336 | 0.629 | 0.332 | 0.599 | 0.373 | 0.605 | 0.472 | 0.653 |
| Traffic 720 | **0.285** | 0.596 | 0.287 | **0.431** | 0.337 | 0.597 | 0.350 | 0.640 | 0.341 | 0.601 | 0.394 | 0.645 | 0.437 | 0.639 |

## 4.2 EXPERIMENT RESULTS

In Table 3, we demonstrate S2GNN's capability in handling both long and short-sequence inputs for spatiotemporal forecasting tasks. Our proposed model not only achieves state-of-the-art prediction performance but also balances training efficiency. Furthermore, in Table 1, following the existing comparison methods for spatiotemporal forecasting and long-sequence prediction tasks, we compare the results of S2GNN with current models used for long-sequence prediction on both spatiotemporal and long sequence datasets.

Moreover, Figure 3 illustrates the shape of the graph filter learned by S2GNN. It can be observed that across all datasets (more results are in Figure 9), the graph filter exhibits a band-rejection filter shape, which strongly indicates that the adaptive graph contains both homophily and heterophily. This observation highlights that current methods using spatial GNNs limit the expressive capacity of the temporal adaptive graph. More specifically, on spatiotemporal datasets, the learned graph filter retains both low-frequency and high-frequency components while suppressing mid-frequency components, suggesting a strong polarization between homogeneity and heterogeneity. Therefore, we suspect that in typical spatiotemporal forecasting tasks, such as traffic flow prediction, ignoring odd-hop neighbors could theoretically enhance the robustness of spatial-GNN models (Lei et al., 2022). On multivariate datasets, however, the presence of mid-frequency components indicates that the polarization in the adaptive graph structure is less pronounced.

## 4.3 ABLATION STUDY

We performed ablation experiments on several datasets to assess the impact of distinguishable features, It can be seen that improving a model's capacity to manage sample indistinguishability can boost its performance on STF. However, the scenario is more intricate with LTSF, possibly due to variations in data properties or evaluation standards.. Additionally, we stopped the gradient updates for the adaptive graph after training for 50 epochs. The results indicate that freezing the adaptive graph had a limited impact on prediction accuracy, suggesting that after a certain number of epochs, further structural changes in the adaptive graph no longer significantly influence the results. This consequence also supports the motivation behind our research.

Table 2: Ablation study. T: Temporal embedding. S: Scale-Adaptive node embedding. C: Cross-Correlaiton embedding. Stop-50: Stop the gradients of the adaptive graph after 50 epochs.

| Methods | PEMS07 | | PEMS08 | | PEMS03 | | Electricity | | Traffic | |
|---------|--------|------|--------|------|--------|------|-------------|------|--------|------|
| Metric | MAE | RMSE | MAE | RMSE | MAE | MSE | MAE | MSE | MAE | MSE |
| w/o T | 27.20 | 44.31 | 19.37 | 30.93 | 21.80 | 34.75 | 0.236 | 0.144 | 0.267 | 0.535 |
| w/o S | 24.61 | 42.11 | 17.64 | 29.43 | 21.26 | 38.05 | 0.373 | 0.139 | 0.252 | 0.506 |
| w/o C | 25.12 | 41.83 | 17.55 | 29.07 | 21.66 | 37.64 | 0.224 | 0.133 | 0.250 | 0.510 |
| stop-50 | 25.46 | 42.18 | 17.55 | 29.03 | 21.94 | 37.96 | 0.224 | 0.134 | 0.251 | 0.509 |
| Ours | 23.70 | 40.50 | 17.36 | 28.88 | 21.24 | 36.96 | 0.226 | 0.135 | 0.251 | 0.508 |

## 4.4 EFFICIENCY STUDY

We compared the number of parameters of existing models with the FLOPs required to train for one epoch. As shown in Figure 1, we selected tasks with different datasets (varying in node scale) and different prediction lengths for comparison. In the figure, the closer a model's point is to the origin, the more worthwhile its computational efficiency. The size of each point reflects the MAE between the predicted results and the true values, where smaller points indicate better performance. It can be observed that S2GNN maintains stable performance across tasks with varying input and output lengths, effectively balancing prediction accuracy and training efficiency.

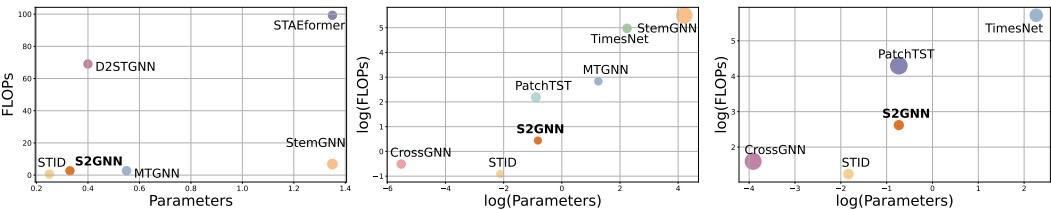

Figure 7: Comparison of FLOPs and total amount of parameters. From left to right are the results on the PEMS04 (12-12), PEMS08 (96-12), and PEMS07 (96-96) datasets. The size of the circles represents the MAE, smaller circles indicate better performance. FLOPs: $\times 10^3$

## 4.5 VISUALIZATION

In Figure 8, we show the results of S2GNN on both spatiotemporal and long-sequence datasets, along with a comparison to three existing models on the same tasks. Due to image size constraints, we selected one node's results from each dataset for illustration. The horizontal axis represents the true values, while the vertical axis shows the predicted values. Ideally, all points should align along the diagonal line $y = x$, indicating perfect predictions.

On spatiotemporal data, MTGNN cannot filter high-frequency information because it relies solely on spatial GNNs, resulting in predictions that are spread on both sides of the diagonal. This issue becomes increasingly evident as the output length increases. CrossGNN, on the other hand,

uses contrastive learning to handle homophilic and heterophilic information on the graph but still struggles with longer sequence forecasting. It tends to underpredict the true values, suggesting difficulty in handling distribution shifts over extended periods. Generally, S2GNN demonstrates a clear advantage in capturing both homophilic and heterophilic relationships in adaptive graphs due to learnable filters. It also maintains stable performance across long-sequence predictions compared with iTransformer, effectively balancing accuracy and adaptability.

On long-sequence data, S2GNN performs slightly worse than spatiotemporal data, possibly because the implicit graph structure in long-sequence data is less pronounced. Additionally, numerical effects introduced by normalization may also impact its performance. Despite this, S2GNN still shows smaller fluctuations compared to the other three models, with data points more tightly clustered around the diagonal. This stability becomes even more pronounced as the prediction length increases.

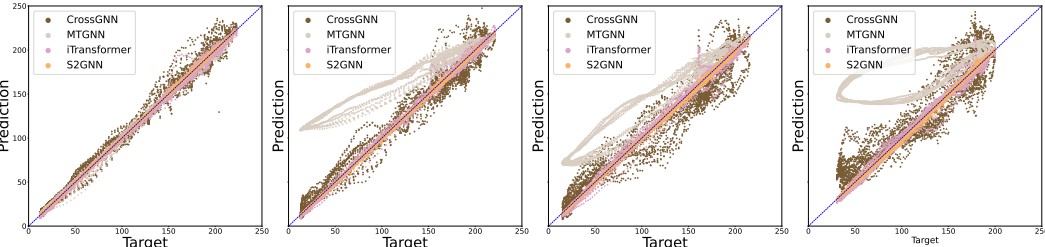

(a) Examples of PEMS04 dataset. We use the 200th node for the sake of simplicity. From left to right, the results depict predictions for 12, 24, 48, and 96 time steps based on an input sequence of 96.

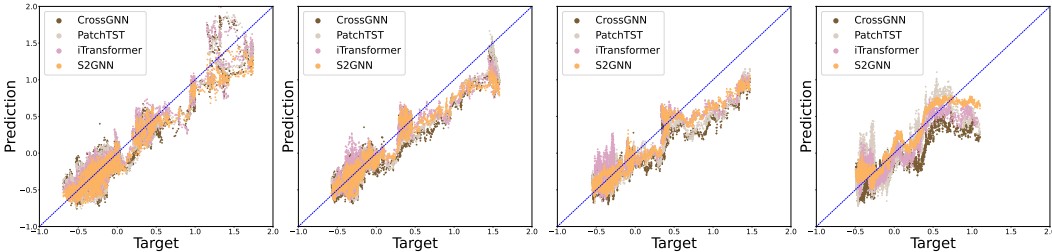

(b) Examples of Elctricity dataset. We use the 60th node for the sake of simplicity. From left to right, the results depict predictions for 96, 192, 336, and 720 time steps based on an input sequence of 96.

Figure 8: Comparison between the prediction results and the target values. Each data point represents the average of the predicted results and the true values over the given sequence.

## 5 CONCLUSION

In this paper, our research is motivated by the inefficiency of existing Spatio-Temporal Graph Neural Networks (STGNNs), which also struggle to handle long-sequence inputs. Through observing the training process of adaptive graphs in existing models, we found that their graph structures tend to stabilize after a few epochs. Based on this, we propose the Spectral Spatio-Temporal Graph Neural Network (S2GNN) to unify short-term and long-term STF while maintaining training efficiency. The decoupled GNN along with an MLP architecture ensures overall efficiency. Specifically, S2GNN incorporates learnable filters that adaptively capture both short-term and long-term features, improving the model's expressiveness. To keep the model simple yet effective, we enhance its ability to handle indistinguishable samples by introducing a Scale-Adaptive node embedding method and a Cross-Correlation embedding method. These methods allow the node embeddings to adjust to varying numbers of nodes and learn distinguishable features directly from input sequences. Experiments on eight public datasets show that our model effectively addresses spatio-temporal prediction for both short and long sequences across different scenarios. It also suggests the importance of modeling both homophilic and heterophilic relationships in MTS forecasting.

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

## A APPENDIX

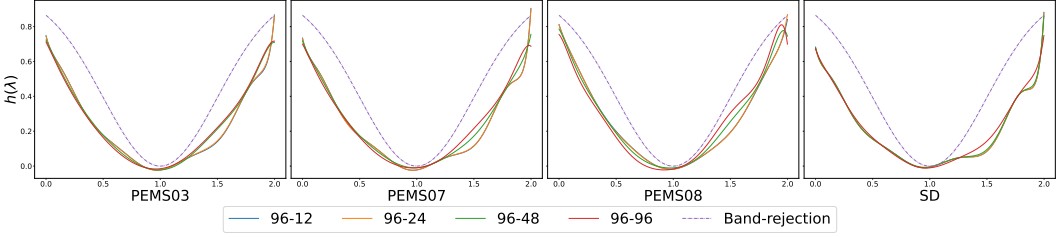

Figure 9: Visualization of learned graph filters. The results show that the filters of the adaptive graph resemble a band-rejection filter across public datasets, indicating the presence of both homophilic and heterophilic relationships between nodes.

Table 3: Comparison of existing models on short-term STF and long-term STF results. Existing models struggle to balance prediction accuracy and training efficiency when handling long-term STF tasks. FLOPs: $\times 10^3$.

| Methods | | Ours | | STID | | D2STGNN | | STAEformer | | MTGNN | | StemGNN | |
|---|---|---|---|---|---|---|---|---|---|---|---|---|---|
| Metric | | 12-12 | 96-12 | 12-12 | 96-12 | 12-12 | 96-12 | 12-12 | 96-12 | 12-12 | 96-12 | 12-12 | 96-12 |
| PEMS03 | MAE | **14.42** | **14.32** | 15.51 | 15.10 | _14.80_ | NA | 14.86 | NA | 14.85 | 14.64 | 16.51 | 57.83 |
| | RMSE | **24.92** | _24.29_ | 27.33 | 26.56 | 25.82 | NA | 25.57 | NA | 25.23 | **23.91** | 27.72 | 83.53 |
| | MAPE | _14.89%_ | _15.29%_ | 16.02% | 16.51% | 15.15% | NA | 15.02% | NA | **14.55%** | **15.20%** | 16.92% | 176.31% |
| | FLOPs | 4.79 | 4.82 | 1.14 | 1.22 | 141 | NA | 184.30 | NA | 4.86 | 52.75 | 12.39 | 759.50 |
| PEMS04 | MAE | **18.05** | **18.16** | 18.22 | _18.31_ | 18.31 | NA | _18.18_ | NA | 19.13 | 18.79 | 24.02 | 72.65 |
| | RMSE | 30.60 | **29.74** | _30.33_ | _29.76_ | **29.82** | NA | 30.24 | NA | 31.03 | 30.55 | 37.04 | 104.81 |
| | MAPE | **11.92%** | **12.51%** | _11.99%_ | _12.53%_ | 12.29% | NA | 12.29% | NA | 13.22% | 13.02% | 17.63% | 54.88% |
| | FLOPs | 2.66 | 2.67 | 0.64 | 0.68 | 69.04 | NA | 99.30 | 832.70 | 2.71 | 29.36 | 6.86 | 422.40 |
| PEMS07 | MAE | **19.03** | **19.10** | 19.31 | _19.44_ | 19.55 | NA | _19.22_ | NA | 21.01 | 20.81 | 22.72 | 75.73 |
| | RMSE | **32.34** | **32.21** | 32.95 | _32.46_ | 32.83 | NA | _32.54_ | NA | 34.14 | 33.83 | 36.99 | 106.81 |
| | MAPE | **8.00%** | **8.08%** | 8.05% | _8.28%_ | 8.23% | NA | _8.03%_ | NA | 8.92% | 9.05% | 9.57% | 82.70% |
| | FLOPs | 13.33 | 13.41 | 3.03 | 3.25 | 884.00 | NA | 660.60 | NA | 12.90 | 140.00 | 35.69 | 2032.00 |
| PEMS08 | MAE | _13.72_ | **13.70** | 14.12 | _14.11_ | 14.20 | NA | **13.60** | NA | 15.25 | 14.93 | 16.37 | 69.20 |
| | RMSE | **23.30** | **22.88** | 23.88 | _23.27_ | _23.65_ | NA | 23.45 | NA | 24.22 | 23.89 | 26.13 | 92.56 |
| | MAPE | _8.98%_ | _9.10%_ | 9.21% | _9.30%_ | 9.26% | NA | **8.94%** | NA | 10.66% | 10.21% | 10.71% | 104.44% |
| | FLOPs | 1.54 | 1.55 | 0.37 | 0.40 | 25.83 | 442.40 | 52.25 | 440.90 | 1.57 | 17.02 | 3.93 | 245.30 |
| SD | MAE | **17.53** | **15.58** | 17.86 | _16.74_ | 17.72 | NA | 18.96 | NA | 18.61 | 17.11 | 23.52 | 74.25 |
| | RMSE | _31.29_ | **26.85** | 31.42 | _29.07_ | **29.91** | NA | 34.89 | NA | 31.57 | 29.50 | 38.05 | 108.82 |
| | MAPE | **11.31%** | **10.11%** | 11.84% | _11.36%_ | 11.89% | NA | 12.26% | NA | 12.81% | 11.84% | 15.98% | 74.55% |
| | FLOPs | 13.16 | 13.25 | 3.05 | 3.28 | 717.40 | NA | 610.20 | NA | 12.99 | 141.20 | 34.92 | 2043.00 |

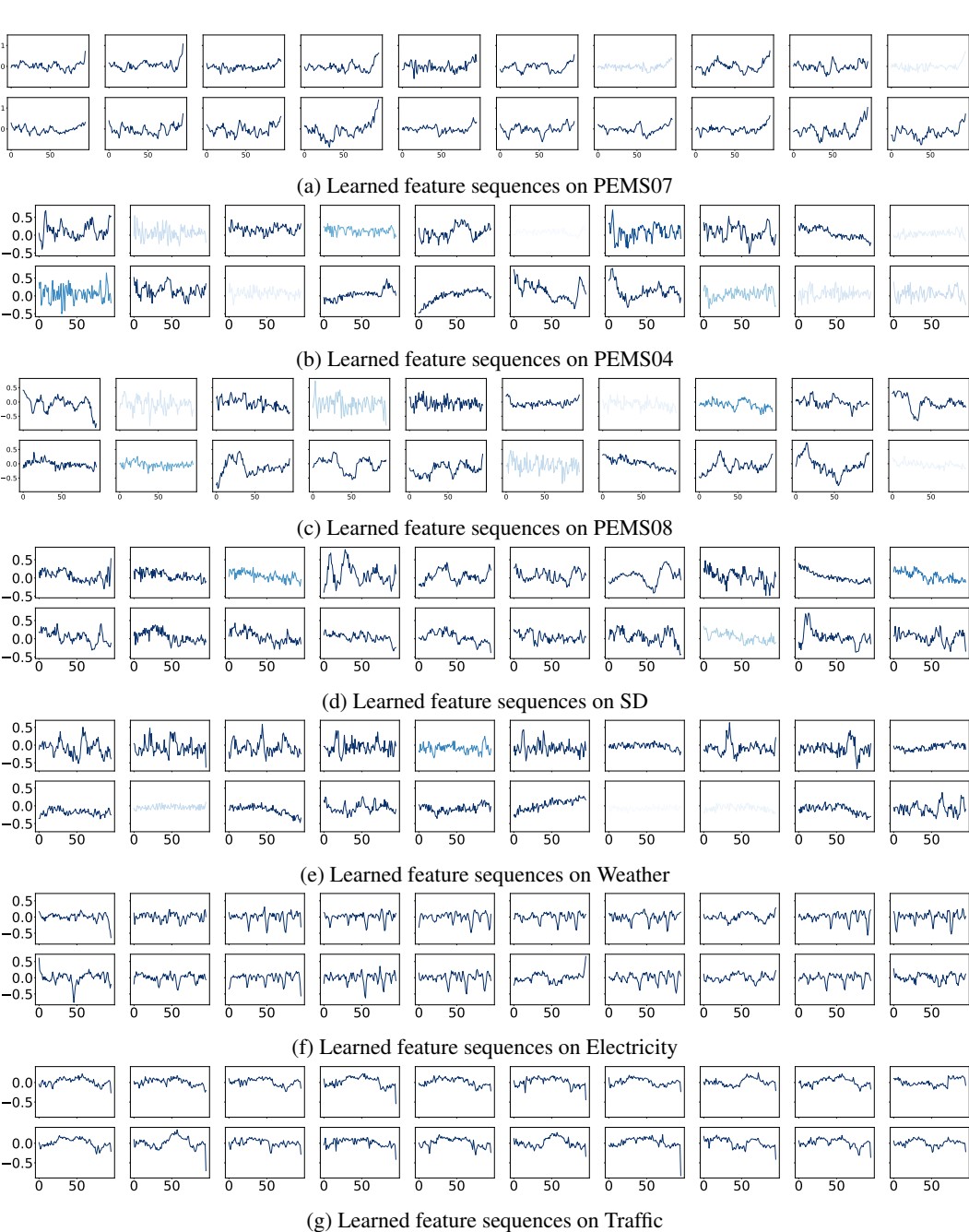

(a) Learned feature sequences on PEMS07

(b) Learned feature sequences on PEMS04

(c) Learned feature sequences on PEMS08

(d) Learned feature sequences on SD

(e) Learned feature sequences on Weather

(f) Learned feature sequences on Electricity

(g) Learned feature sequences on Traffic

Figure 10: More visualizations of the convolutional kernels used in the cross-correlation embedding process, where the color intensity represents their usage frequency (i.e., the number of times each kernel is connected).

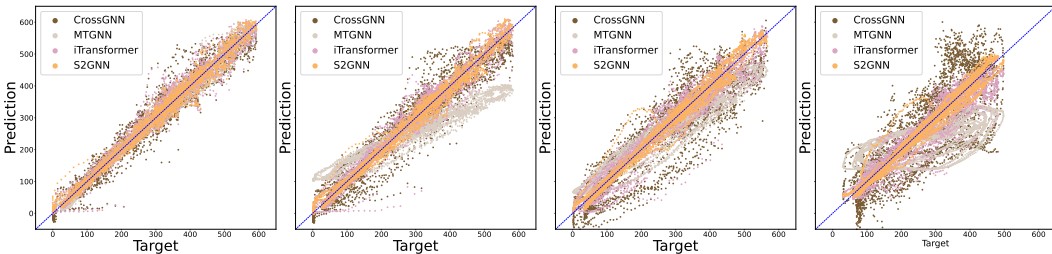

(a) More Eexamples of PEMS04 dataset. We use the 201st node for the sake of simplicity. From left to right, the results depict predictions for 12, 24, 48, and 96 time steps based on an input sequence of 96.

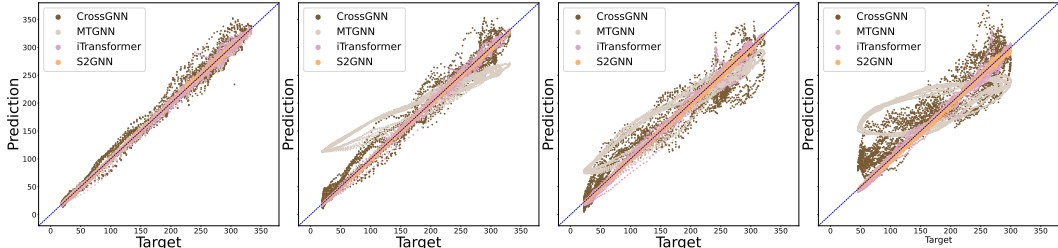

(b) More Eexamples of PEMS04 dataset. We use the 150th node for the sake of simplicity. From left to right, the results depict predictions for 12, 24, 48, and 96 time steps based on an input sequence of 96.

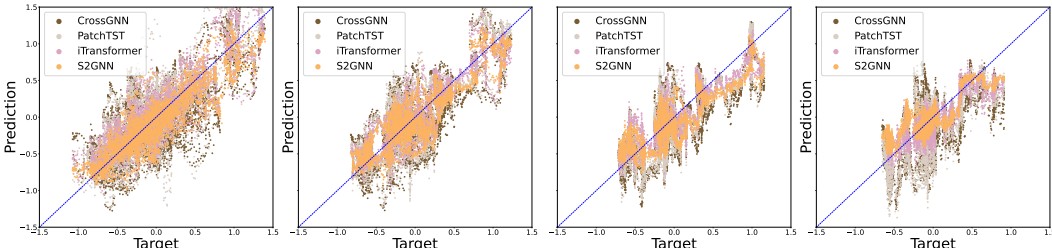

(c) More examples of Elctricity dataset. We use the 50th node for the sake of simplicity. From left to right, the results depict predictions for 96, 192, 336, and 720 time steps based on an input sequence of 96.

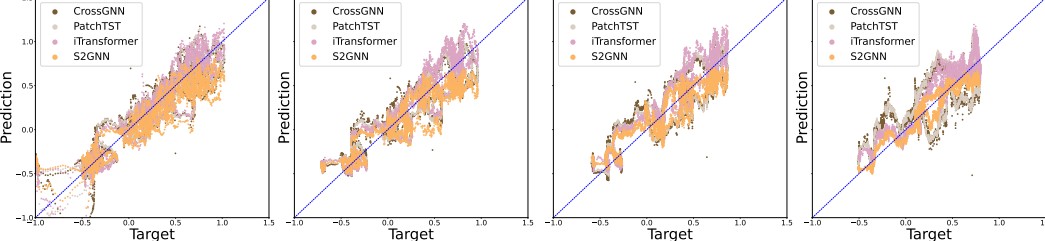

(d) More examples of Elctricity dataset. We use the 150th node for the sake of simplicity. From left to right, the results depict predictions for 96, 192, 336, and 720 time steps based on an input sequence of 96.

Figure 11: More comparison between the prediction results and the target values. Each data point represents the average of the predicted results and the true values over the given sequence.

