# OpenReview forum: "UNIFYING LONG AND SHORT SPATIO-TEMPORAL FORECASTING WITH SPECTRAL GRAPH NEURAL NETWORKS"
_ICLR.cc/2025/Conference — Submitted to ICLR 2025_

### Official Review · Reviewer_zKgV · 2024-10-27

**Soundness:** 3
**Presentation:** 2
**Contribution:** 2
**Rating:** 6
**Confidence:** 4

**Summary:**

This paper introduces a novel model, the Spectral Spatio-Temporal Graph Neural Network (S2GNN), aimed at unifying short-term and long-term multivariate time series forecasting. Current research in this area is primarily divided into Spatio-Temporal Forecasting (STF) and Long-term Time Series Forecasting (LTSF), which differ significantly in methods and benchmarks. S2GNN leverages the strengths of spectral graph neural networks and incorporates an adaptive graph structure to efficiently handle varying sequence lengths. By introducing scale-adaptive node embeddings and cross-correlation embeddings, S2GNN demonstrates superior performance across multiple public datasets, outperforming state-of-the-art models.

**Strengths:**

1  The proposed S2GNN model effectively combines the advantages of short-term and long-term forecasting, addressing a significant gap between STF and LTSF research.
2  Extensive experiments on eight public datasets show that S2GNN consistently outperforms existing models, indicating strong generalization capabilities.
3  The use of an adaptive graph structure and learnable filters allows the model to flexibly accommodate different input lengths, enhancing its expressiveness.
4  The paper clearly outlines the model's construction and its components' functionality, making it accessible for other researchers to understand and replicate.

**Weaknesses:**

1  While S2GNN performs well across various datasets, its computational complexity may pose challenges in training on larger datasets.
2  Despite introducing learnable filters, the internal mechanisms and decision-making processes of the model lack sufficient interpretability, which may affect trust in practical applications.
3  The selection of baseline models for comparison is somewhat limited, potentially failing to fully showcase the advantages of S2GNN.

**Questions:**

NAN

---

> ### Author Response · Authors · 2024-11-24
> **Request of Reviewer's feedback**
>
> Dear Reviewer,
>
> We’d like to make sure we’ve fully answered your questions. If there’s anything else you need clarification on, please feel free to let us know. Thanks again for your time!

---

> ### Author Response · Authors · 2024-12-03
>
> Thank you for taking the time to review our paper. As the discussion period is approaching its end, we would like  to check whether our responses have addressed your questions.
>
> We have revised the manuscript, including improvements to the figures, adding iTransformer as a baseline model, and modifying several sentences to improve the overall coherence of the manuscript.
>
> We hope our responses can adequately address the your concerns and questions.

---

### Official Review · Reviewer_ejuH · 2024-11-02

**Soundness:** 2
**Presentation:** 1
**Contribution:** 2
**Rating:** 5
**Confidence:** 4

**Summary:**

The authors aimed to achieve efficient learning for GNNs to process both short- and long-term spatial-temporal dependencies for forecasting tasks. To achieve so, they proposed S2GNN, introducing learnable filters to process these dependencies.

**Strengths:**

1.	The addressed problem, achieving efficient learning while simultaneously capturing spatial-temporal information, is important.
2.	The experiments are comprehensive, including multiple datasets for comparisons with SOTAs, ablation study, efficiency study etc.

**Weaknesses:**

1.	Motivation is unclear. I think the main motivation should be achieving efficient learning and meanwhile capturing the spatial-temporal information. However, the authors discuss a lot regarding the challenges in sample indistinguishability, which really confuse me. The authors should explain, why you adopted sample indistinguishability? what is its function in achieving training efficiency? Additionally, why adaptive graph structure is used? Still, what is its function in achieving training efficiency?
2.	Contribution is insufficient and not novel enough. The contributions include scale-adaptive node embeddings and cross-correlation embeddings. (I don’t think the adaptive adjacent matrix is a contribution, because it just employs a trainable weight to each node.) I appreciate the contribution of the scale-adaptive node embeddings, which is interesting, although I don’t see its connection with RIP. But the contribution is not enough for ICLR. Regarding the cross-correlation embeddings, I don’t see how it can help to achieve more efficient learning to process both short and long term dependencies.
3.	Writing is quite poor. The manuscript does not sound logical. For example, the authors discuss a lot the pros of spatial GNN and the cons of spectral GNN, and then they adopted spectral GNN in this work. I dont think this is logical.

**Questions:**

See above

---

> ### Author Response · Authors · 2024-11-24
> **Request of Reviewer's feedback**
>
> Dear Reviewer,
>
> We’d like to ensure that we’ve answered all of your questions. Please let us know if our response has resolved your concerns, or if you require any additional clarification. Thank you for your time!

---

> > ### Comment · Reviewer_ejuH · 2024-11-28
> >
> > I appreciate the authors' response. I can raise my score but I still dont think it meets the standard of ICLR, including the technique contribution and writing.

---

> > > ### Author Response · Authors · 2024-11-28
> > > **Response to Reviewer ejuH**
> > >
> > > Thank you for your positive feedback and valuable suggestions. Our research proposes S2GNN from the perspective of spatiotemporal prediction. This method adopts a decoupled + MLP-based architecture, which structurally improves the efficiency of spatiotemporal prediction. Additionally, enhancing the distinguishability of samples provides a novel approach to solving spatiotemporal prediction problems (see Table 1 below). This approach **can handle both short-input and long-sequence prediction tasks** and has demonstrated state-of-the-art performance on several long-sequence prediction datasets.
> > >
> > > We have **updated Figures 1, 2, and 8**, and added several sentences to improve the overall coherence of the manuscript (**Marked as blue**).  We will further review the entire manuscript to enhance clarity and accessibility.
> > >
> > > Table 1 The impact of enhanced distinguishability.
> > > |      |           | PEMS04 |      | PEMS08 |      | Weather |     | Electricity | |
> > > |:---|:---|:---|:---|:---|:---|:---|:---|:---|:---|
> > > |Model       | Horizon   | MAE   | RMSE  | MAE   | RMSE  | MAE   | MSE   | MAE   | MSE   |
> > > |Mlp         | 96-12/96  | 21.89 | 34.76 | 16.37 | 26.42 | 0.211 | 0.176 | 0.267 | 0.191 |
> > > |Mlp+embs    | 96-12/96  | 18.31 | 29.76 | 14.11 | 23.27 | 0.203 | 0.161 | 0.267 | 0.188 |
> > > |S2GNN       | 96-12/96  | 18.16 | 29.74 | 13.70 | 22.88 | 0.198 | 0.154 | 0.226 | 0.135 |

---

> ### Author Response · Authors · 2024-12-03
>
> Thank you for taking the time to review our paper. As the discussion period is approaching its end, we hope that you will reconsider our work for the following reasons:
> - The STGNN model excels at handling the relationships between variables in sequential data but generally suffers from inefficiency. We propose a solution to this issue by removing inter-layer weights to improve efficiency.
> - Furthermore, most existing approaches rely on the homophily assumption, such as using similarity measures based on attention mechanisms to assess the influence between variables. In contrast, our work directly demonstrates the existence of heterophily relationships between temporal variables by visualizing the weights of filters, highlighting that considering only homophily relationships is insufficient.
> - In the revised manuscript, we emphasize that our work structurally ensures an improvement in spatiotemporal prediction efficiency, and we have also made adjustments to the figures in the paper.
>
> We again thank you for your insightful review and suggestions to improve our paper.

---

### Official Review · Reviewer_kyrF · 2024-11-02

**Soundness:** 1
**Presentation:** 1
**Contribution:** 1
**Rating:** 3
**Confidence:** 5

**Summary:**

This paper focuses on **unifying long and short spatio-temporal forecasting**. The main research contribution lies in the design of **input embeddings**, introducing a **scale-adaptive node embedding method** and a **cross-correlation embedding method**. Experiments were conducted on both **STF datasets** and **LTSF datasets**. However, the overall motivation is not clear, the method lacks innovation, and the experiments fail to convincingly demonstrate that the approach is SOTA.

**Strengths:**

The main drawback of the STF method in the author's motivation is that it requires a lot of computing resources and the model is inefficient when performing LTSF tasks. As far as I know, such a problem does exist. The main research contribution lies in the design of **input embeddings**, introducing a **scale-adaptive node embedding method** and a **cross-correlation embedding method**.

**Weaknesses:**

The overall motivation is not clear, the method lacks innovation, and the experiments fail to convincingly demonstrate that the approach is SOTA.

**Motivation:** The author's motivation is that performing long-term spatio-temporal forecasting (LTSF) tasks requires substantial computational resources, and the model is inefficient. The author attempts to enhance model efficiency through sample indistinguishability, but the connection between sample indistinguishability and model efficiency is not clearly explained.

**Originality:** The primary contribution lies in the design of embeddings, specifically a **scale-adaptive node embedding method** and a **cross-correlation embedding method**. However, the distinction between these embeddings and temporal or positional embeddings is unclear. As stated in Section 3.1, "This value could be the node’s ID, the time of the day, or the day of the week." if the cross-correlation embedding method is a typical temporal embedding with an added node‘ ID？

**Quality:** The writing quality is average, with several errors (e.g., in the explanation of Equation 2, "Where c is a constant constant," there is a redundant "constant"). The figures are not sufficiently clear, with dimensions that are too small, and the explanations and labels for all figures lack clarity.

**Questions:**

0.  My main concern is: The size of model does not appear to be related to the embedding design, as it is likely the GNN component that primarily influences model size. So **what is the connection between sample indistinguishability and model efficiency or size?**  Please provide a more detailed explanation of how addressing sample indistinguishability specifically contributes to improved model efficiency or reduced computational requirements.


1. **Please use figures to explain your statement that "existing methods for handling spatial distinguishability cannot adapt to changes in the number of nodes."**
   - 1) Is your model inductive?
   - 2) How does S2GNN handle changes in the number of nodes?
   - 3) What types of node changes are you addressing? If the number of nodes in the test dataset exceeds that of the training dataset, can your model still make predictions accurately? Can it handle cases of missing or additional nodes?

2. **As most TSF benchmarks achieve good performance within 10 epochs, why does your model require so many epochs, as shown in Figure 2?** Additionally, could you explain why the graph structure appears stable and lacks significant variation? Is it possible that this is simply the result of model convergence?

3. **As stated in Section 3.1, "This value could be the node’s ID, the time of the day, or the day of the week," is the cross-correlation embedding method just a typical temporal embedding with an added node ID?** Please provide a more detailed comparison between the proposed embedding methods and existing temporal/positional embeddings, highlighting the key differences and advantages. This would help clarify the novelty of S2GNN.

4. **I don't see any components designed specifically to optimize both short- and long-term TSF tasks.** Could you concisely explain why your model qualifies as a unified model for both short- and long-term TSF?

5. **The chosen baselines are not previous SOTA, which limits the credibility of your experimental comparisons.** Please add recent SOTA baselines, e.g. itransformer, forecastgrapher.

---

> ### Author Response · Authors · 2024-11-20
> **Rebuttal by Authors (Part 1)**
>
> We thank the reviewer for offering the valuable feedback. We have addressed each of the concerns raised by the reviewer as outlined below.
>
> **W1. Motivation**
>
> The efficiency improvement of S2GNN primarily comes from its adoption of a decoupled GNN structure, which eliminates the need for inter-layer parameter matrices, thereby reducing computational complexity. Additionally, by addressing sample indistinguishability, we can use an MLP-based model instead of the commonly used complex sequence structures. This substitution maintains predictive accuracy while significantly lowering computational costs.
>
> **W2. Originality**
>
> Our primary contribution lies in utilizing a **decoupled GNN + MLP-based** structure to address the efficiency issues common in most existing spatiotemporal prediction models. This novel approach provides an efficient alternative for STGNN-type models to handle long input sequences, offering significant advantages over traditional **GCN+CNN-based, GCN+RNN-based, or attention-based models.** S2GNN is not only structurally more efficient but also maintains strong performance for both short-sequence and long-sequence spatiotemporal forecasting tasks.
>
> Regarding the embeddings, the cross-correlation embedding is not simply a typical temporal embedding with an added node ID. Instead, all the feature embedding $H$ of the input sequence is obtained through matrix indexing, where $W$ is the feature matrix and $X$ is the input sequence. Specifically, the feature embedding is computed as:
> $H = W[f(X)]$,
> where $f(X)$ represents the indexing function applied to the input sequence $X$. The distinction between various feature embeddings lies in the method of computing this index. In the case of **cross-correlation embedding**, the index is derived from the similarity between the input sequence and multiple convolutional kernels. On the other hand, the indices for time embeddings and node embeddings are based on the inherent information present within the input sequence itself.
>
> **W3. Quality**
>
> Thank you for pointing out these issues. We promise to thoroughly review and revise the manuscript in the next stage.
>
> **Q0.**
>
> The improvement in model efficiency comes from employing **a decoupled GNN + MLP-based** architecture. This combination reduces computational requirements by eliminating inter-layer parameter matrices, making the model more scalable and efficient.
>
> Addressing sample indistinguishability is crucial primarily for improving accuracy. By enhancing the model's ability to distinguish between similar samples, we can maintain high predictive performance even with a simpler MLP-based model. This is significant because it allows us to replace the commonly used, computationally expensive sequence structures (such as RNNs or attention mechanisms) with a more efficient MLP-based model. As a result, we reduce the overall computational cost while still achieving strong accuracy in spatiotemporal prediction tasks.
>
> **Q1.**
>
> The variation in the number of nodes refers to different datasets having different numbers of nodes. In existing STGNN models, fixed-size embeddings are used to learn node similarity, which becomes problematic when the number of nodes varies across datasets. For example,  PEMS08 has 170 nodes, while PEMS07 has 883 nodes, yet the model uses the same embedding size for both datasets. This inconsistency makes it difficult to adapt the embeddings efficiently. Our proposed method mitigates this issue. For instance, assuming $c = 10$, $ r = 0.5 $, and $ d = 32$ , existing methods need to adjust the distances between 170 and 883 vectors of length 32 in the feature space. In contrast, our approach only requires handling 29 and 37 vectors, respectively, making adaptation more efficient.
> Therefore, our model is inductive and designed to handle variations in the number of nodes across different datasets. However, scenarios involving missing or additional nodes within the same dataset, or when the number of nodes in the test dataset exceeds that in the training dataset, are outside the scope of this work.

---

> ### Author Response · Authors · 2024-11-20
> **Rebuttal by Authors (Part 2)**
>
> **Q2.**
>
> We believe that many factors determine the convergence speed of a model, such as the learning rate, dataset size, and the influence of random seeds. While some transformer-based models converge in fewer epochs, they often require more time per training epoch. In contrast, our method may need more epochs to converge, but trains faster per epoch, resulting in a reasonable total running time, as shown in Table 1. Note that although Fredformer converges in fewer epochs, its performance on spatiotemporal prediction tasks is suboptimal, as highlighted in Table 3 of Q5.
>
> Additionally, Figure 2 illustrates the structural changes in the adaptive graphs of two existing models that use adaptive graph structures, not our proposed model. The stabilization of the adaptive graph structure indicates that the GNN part of the model has converged. We describe this from the perspective of graph structure to explain that applying spectral graph models to adaptive graphs is feasible.
>
> **Table 1** Comparison of training time. Best refers to the epoch when val loss is minimum.
>
> |               |          | PEMS04 |         | PEMS08 |         | Weather |         | Electricity |    |
> |:---|:---|:---|:---|:---|:---|:---|:---|:---|:---|
> |Model          | Horizon  | Time/Epoch | Best | Time/Epoch | Best |  Time/Epoch | Best | Time/Epoch | Best |
> |iTransformer   | 96-12/96 | 12.75s     | 98  | 7.13s      | 100 | 7.13s       | 100 | 26.63s     | 76  |
> |Fredformer     | 96-12/96 | 20.28s     | 11  | 8.99s      | 11  | 8.65s       | 12  | 53.91s     | 40  |
> |ForecastGrapher| 96-12/96 | 191.62s    | NA  | 112.61s    | NA  | 48.74s      | 2   | 380.12s       | NA  |
> |S2GNN          | 96-12/96 | 9.74s      | 65  | 4.38s      | 91  | 21.02s      | 10  | 24.32s     | 96  |
>
> **Q3.**
>
> I’m not entirely sure what you mean by "a typical temporal embedding with an added node ID." To clarify, we are not adding node IDs to the input sequence redundantly.
>
> Simply speaking, sample indistinguishability occurs when dynamics from different factors overlap, making input sequence features too similar to distinguish. To address this, we feed the model with more information to help distinguish these inputs. The essence of node IDs or temporal features is to provide an index for the input sequence and then concatenate a feature embedding vector to the input sequence based on the index value.
>
> In our method, we compute the cross-correlation between the input sequence and a set of feature sequences to derive an index value. Then, we concatenate the corresponding feature embedding vector to the input vector. Figure 5 shows what these feature sequences look like ultimately. Specifically, we calculate the similarity between the input sequence and these feature sequences, assign the feature embedding vector corresponding to the most similar feature sequence, and concatenate it with the input vector. This does not involve directly concatenating the feature sequence itself.
>
> This process is the same as how the previous three features (Node ID, time of the day, day of the week) are handled.
>
> **Q4**
>
> Addressing sample indistinguishability is important to maintaining performance across varying forecasting horizons. As demonstrated in [1], it is effective for enhancing TSF outcomes. We extend this by strengthening the model's ability to handle inter-variable relationships, allowing it to excel in both short- and long-term forecasting tasks.
>
> To illustrate this, we conducted an experiment comparing three models: the base MLP, MLP+embs, and our full S2GNN model. As shown in Table 2, our full S2GNN model achieves the best results, emphasizing that integrating embeddings, alongside the decoupled GNN architecture, is crucial for achieving superior LTSF performance.
>
> **Table 2** The impact of enhanced distinguishability.
>
> |      |           | PEMS04 |      | PEMS08 |      | Weather |     | Electricity | |
> |:---|:---|:---|:---|:---|:---|:---|:---|:---|:---|
> |Model       | Horizon   | MAE   | RMSE  | MAE   | RMSE  | MAE   | MSE   | MAE   | MSE   |
> |Mlp         | 96-12/96  | 21.89 | 34.76 | 16.37 | 26.42 | 0.211 | 0.176 | 0.267 | 0.191 |
> |Mlp+embs    | 96-12/96  | 18.31 | 29.76 | 14.11 | 23.27 | 0.203 | 0.161 | 0.267 | 0.188 |
> |S2GNN       | 96-12/96  | 18.16 | 29.74 | 13.70 | 22.88 | 0.198 | 0.154 | 0.226 | 0.135 |
>
> 【1】Spatial-Temporal Identity: A Simple yet Effective Baseline for Multivariate Time Series Forecasting，CIKM 2022（short）

---

> ### Author Response · Authors · 2024-11-20
> **Rebuttal by Authors (Part 3)**
>
> **Q5** More baselines
>
> To ensure a fair comparison, we implemented our method and all baselines on the open-source platform BasicTS with a batch size equal to 64. Incorporating feedback from other reviewers, we include additional baselines (iTransformer [2] and Fredformer [3]) to show our approach's competitiveness (as shown in Table 3). Due to out-of-memory (OOM) errors, we only report ForecastGrapher[4]’s results for the 96-96 task on the Weather dataset.
>
> **Table 3.** More baselines.
> |               |           | PEMS04 |      | PEMS08 |      | Weather |     | Electricity | |
> |:---|:---|:---|:---|:---|:---|:---|:---|:---|:---|
> |Model          | Horizon   | MAE   | RMSE  | MAE   | RMSE  | MAE   | MSE   | MAE   | MSE   |
> |iTransformer   | 96-12/96  | 20.28 | 32.38 | 15.13 | 24.72 | 0.217 | 0.175 | 0.236 | 0.147 |
> |iTransformer   | 96-96/720 | 30.38 | 47.55 | 23.46 | 38.42 | 0.353 | 0.356 | 0.301 | 0.216 |
> |Fredformer     | 96-12/96  | 23.66 | 36.82 | 18.59 | 29.20 | 0.194 | 0.157 | 0.230 | 0.139 |
> |Fredformer     | 96-96/720 | 38.88 | 57.17 | 33.56 | 50.05 | 0.335 | 0.342 | 0.297 | 0.215 |
> |ForecastGrapher| 96-12/96  | OOM   | OOM   | OOM   | OOM   | 0.197 | 0.158 | OOM   | OOM   |
> |S2GNN          | 96-12/96  | 18.16 | 29.74 | 13.70 | 22.88 | 0.198 | 0.154 | 0.226 | 0.135 |
> |S2GNN          | 96-96/720 | 21.47 | 34.63 | 17.36 | 28.88 | 0.363 | 0.366 | 0.305 | 0.215 |
>
> 【2】iTransformer: Inverted Transformers Are Effective for Time Series Forecasting， ICLR 2024
>
> 【3】Fredformer: Frequency Debiased Transformer for Time Series Forecasting， KDD 2024
>
> 【4】ForecastGrapher: Redefining Multivariate Time Series Forecasting with Graph Neural Networks， arXiv 2024

---

> ### Comment · Reviewer_kyrF · 2024-11-21
>
> I regret that the quality of this work is not satisfactory. Many essential revisions to the manuscript will need to wait until the next stage. Additionally, the reliability of the proposed S2GNN has not been compared with recent SOTA in the revised paper. The presentation of the work is also poor. I apologize to the authors for the time and effort invested in preparing the rebuttal.
>
> Consequently, I am inclined to reject the submission and I will maintain my current score.

---

> ### Author Response · Authors · 2024-11-21
> **Response to Reviewer kyrF**
>
> I appreciate your review and feedback. However, could you please detail the necessary changes and clarify which parts of the manuscript were unclear or poorly presented? Additionally, we have included the results of iTransformer and ForecastGrapher. It is evident that their performance is not satisfactory in terms of efficiency or accuracy.

---

> ### Author Response · Authors · 2024-11-22
> **Response to Reviewer kyrF**
>
> **R1**
>
> We respectfully disagree with the assessment that S2GNN merely optimizes the input embedding. S2GNN employs a **decoupled GNN + MLP-based architecture**, which effectively addresses the challenge of handling long input sequences in spatiotemporal forecasting tasks. We believe that the value of a novel method lies not in its complexity but in its simplicity and effectiveness.
>
> **R2**
>
> Thank you for your advice. We have updated Figures 1 and 2 in the revised PDF, enlarging the text to improve readability and ensure better visualization. If you have any further suggestions, please let us know.
>
> **R3**
>
> We have conducted extensive experiments, including evaluations on eight real-world datasets, to demonstrate S2GNN's capabilities in handling both short- and long-term spatiotemporal forecasting tasks. Additionally, we performed ablation studies to validate our design choices. Considering feedback from all reviewers, we will include more baselines and add comparisons of training efficiency in the next version to provide a more comprehensive evaluation. We believe these experiments sufficiently demonstrate the superior performance of S2GNN, **especially in spatiotemporal forecasting** tasks. If you have further suggestions for additional experiments, please let us know.
>
> **R4**
>
> Thank you for your feedback. To ensure fair comparisons and reproducible results, we implemented our method using the **open-source platform BasicTS**. If you have trouble reproducing the results in our paper, we can offer assistance.

---

> > ### Author Response · Authors · 2024-11-24
> > **Request of Reviewer's feedback**
> >
> > Dear Reviewer,
> >
> > We want to confirm whether we’ve fully addressed your questions. Please let us know if our response has resolved your concerns, or if you need any further clarification. Thanks for your time!

---

> ### Author Response · Authors · 2024-12-03
>
> Thank you for taking the time to review our paper. As the discussion period is approaching its end, we would like to confirm whether our responses have adequately addressed your questions. In response to your comments, we have:
> - optimized most of the figures in the manuscript and added more examples in the appendix.
> - added iTransformer as a baseline model and included more ablation studies to demonstrate the effectiveness of our proposed approach.
> - improved the overall coherence of the manuscript in the revised PDF.
>
> Thank you once again for your valuable comments and suggestions to improve our paper.

---

### Official Review · Reviewer_RbR3 · 2024-11-03

**Soundness:** 3
**Presentation:** 3
**Contribution:** 3
**Rating:** 8
**Confidence:** 5

**Summary:**

This paper presents a spatiotemporal graph model composed of learnable graph filters and an MLP-based architecture, addressing the efficiency challenges faced by existing spatiotemporal graph models when processing long-sequence inputs. In addition, the authors propose two feature embedding methods that enhance the model's capability to handle data indistinguishability issues. Moreover, using spectral GNN with learnable filters, the author shows that low-pass and high-pass information must be considered when utilizing the adaptive graph in STF. Experimental results on eight datasets reveal that the model outperforms state-of-the-art models, achieving superior predictive capability while maintaining reasonable training efficiency.

**Strengths:**

1. The paper provides a comprehensive and clear definition of the studied problems.
2. The paper is well-structured and easy to follow.
3. The use of a spectral GNN framework and a simple MLP-based sequential model ensures that the model remains scalable and efficient, making it applicable to longer input as well as larger datasets.
4. The experiments cover a wide range of public spatial-temporal and long-term time series datasets, showcasing the robustness and effectiveness of the proposed method.
5. The use of learnable filters enhances the interpretability of the model, directly demonstrating that temporal data exhibit both homophily and heterophily.
6. The scalability and efficiency problems are significant and practical for real-world traffic forecasting applications.

**Weaknesses:**

1. The font is too small in Figures 2, making it hard to read.
2. The paper lacks a comparison with other benchmark models in LTSF, such as the iTransformer[1].
3. The paper lacks evaluation of large-scale spatiotemporal datasets and comparisons with models specifically designed for such datasets, such as BigST[2].

[1] iTransformer: Inverted Transformers Are Effective for Time Series Forecasting

[2] BigST: Linear Complexity Spatio-Temporal Graph Neural Network for Traffic Forecasting on Large-Scale Road Networks

**Questions:**

See weakness

---

> ### Comment · Reviewer_RbR3 · 2024-11-25
> **Thanks for the rebuttal.**
>
> My concerns are addressed, and I will maintain my score and confidence.

---

> > ### Author Response · Authors · 2024-12-03
> >
> > As the discussion period is approaching its end, we would like thank you for your positive support once again.

---

### Official Review · Reviewer_uMZs · 2024-11-12

**Soundness:** 2
**Presentation:** 3
**Contribution:** 2
**Rating:** 5
**Confidence:** 4

**Summary:**

This paper presents a method to integrate spatiao-temporal with long-term time series forecasting, tasks which were previously at odds with one another in terms of model capabilities (spatio-temporal models were usually unable to deal with long-term forecasting, while long-term forecasters were inherently incapable of handling complex variable relationships). The authors propose S2GNN (Spectral Spatio-Temporal Graph Neural Network), which unifies both approaches by the following innovations:

1. Using spectral GNNs with learnable filters
2. Incorporating scale-adaptive node embeddings
3. Utilizing cross-correlation embeddings
4. Adopting an adaptive graph structure

Extensive experiments show SOTA performance over baseline models

**Strengths:**

1. This paper solves an existing and urgent problem in time series forecasting by combining long-term and spatio-temporal forecasting;
2. It utilizes novel model designs which reduces the length requirement on the input training sequence (is this the purported reason for its utility in LTSF?)
3. Experiments generally show SOTA performance across a variety of metrics, although it is unclear how this model outperforms baselines on LTSF;

**Weaknesses:**

1. In the methodology section, it is not made explicit where the ability of the model to model long-term time series comes from, and how;
2. Correspondingly, in the Experiments section, specifically in Fig. 8(b), visually, the performance of the model on long-term time series does not seem to be much better than baseline models, although the authors claim that the results show smaller fluctuations around the diagonal;
3. In the Experiments section, to show superiority wrt long-term time series, the authors should compare with models that have been designed to deal with long-term time series; an example could have been the xLSTM;
4. Ablation studies also do not quite illustrate which module is particularly responsible for utility on LTSF;

**Questions:**

1. Can the authors point out explicitly which part in the model design is responsible for the good performance on LTSF?
2. Why is the visualization only provided for two other baseline models?
3. Which part (ablation study) is responsible for LSTF?

---

> ### Author Response · Authors · 2024-11-24
> **Request of Reviewer's feedback**
>
> Dear Reviewer,
>
> We have added additional visual results similar to those in Figure 8 in the appendix (Figure 11). In addition, we promise to include iTransformer as a baseline model and provide a visualization of its results for comparison. As the discussion period is nearing its conclusion, we welcome any further comments on our rebuttal or the revised paper. Thank you for your time!

---

> ### Author Response · Authors · 2024-12-03
>
> Thank you for taking the time to review our paper. As the discussion period is approaching its end, we would like to check whether our responses have addressed your questions. Following your comments, we:
>
> - We have updated Figures 1, 2, and 8, and added several sentences to improve the overall coherence of the manuscript.
> - In our ablation study (Table 2), we provide more feature embeddings that relate to the model's predictive performance, further emphasizing the necessity of handling indistinguishability among samples.
> - We have included iTransformer as a baseline model and visualized the results in Figure 8.
>
> Thank you again for your valuable comments and suggestions to improve our paper. We look forward to your reply.

---

### Meta-Review · Area_Chair_5CKM · 2024-12-22

**Metareview:**

This paper introduces a model, the Spectral Spatio-Temporal Graph Neural Network (S2GNN), aiming to bridge the gap between short-term and long-term multivariate time series forecasting. However, there are several critical concerns about this paper, such as lack of clear motivation, insufficient novelty, and severe issues with presentation/writing. All the reviewers pointed out the writing issues. In addition, the proposed technical contributions, including scale-adaptive node embeddings, cross-correlation embeddings, and an adaptive adjacency matrix, lack substantial originality. For these limitations, I would like to recommend rejecting this paper.

**Additional Comments On Reviewer Discussion:**

During the rebuttal period, 4 out 5 reviewers responded to the authors’ replies. Reviewer RbR3 and Reviewer zKgV maintain their positive scores. Reviewer kyrF also kept the original negative score due to unsatisfactory technical quality and poor presentation. Reviewer ejuH increased the score, but this reviewer still thinks this paper is below the bar of ICLR due to the technical contribution and writing issues. Reviewer uMZs did not respond to the authors’ replies. But I think the comment about the motivation of this work (i.e., the connection between LTSF performance and sample indistinguishability) is still not clear. Overall, I agree with Reviewer ejuH and Reviewer kyrF about the technical contrition and writing issues.

---

### Decision · Program_Chairs · 2025-01-22

Reject